# Identification and Validation of Candidate Genes Conferring Resistance to Downy Mildew in Maize (*Zea mays* L.)

**DOI:** 10.3390/genes11020191

**Published:** 2020-02-11

**Authors:** Hyo Chul Kim, Kyung-Hee Kim, Kitae Song, Jae Yoon Kim, Byung-Moo Lee

**Affiliations:** 1Department of Life Science, Dongguk University-Seoul, Seoul 04620, Korea; hyocherry@dongguk.edu (H.C.K.); redanan@dongguk.edu (K.-H.K.); skt526@gmail.com (K.S.); 2Department of Plant Resources, College of Industrial Science, Kongju National University, Yesan 32439, Korea; jaeyoonkim@kongju.ac.kr

**Keywords:** maize, downy mildew, QTL, qRT-PCR, RILs

## Abstract

Downy mildew (DM) is a major disease of maize that causes significant yield loss in subtropical and tropical regions around the world. A variety of DM strains have been reported, and the resistance to them is polygenically controlled. In this study, we analyzed the quantitative trait loci (QTLs) involved in resistance to *Peronosclerospora sorghi* (sorghum DM), *P. maydis* (Java DM)*, and Sclerophthora macrospora* (crazy top DM) using a recombinant inbred line (RIL) from a cross between B73 (susceptible) and Ki11 (resistant), and the candidate genes for *P. sorghi*, *P. maydis*, and *S. macrospora* resistance were discovered. The linkage map was constructed with 234 simple sequence repeat (SSR) and restriction fragment length polymorphism (RFLP) markers, which was identified seven QTLs (chromosomes 2, 3, 6, and 9) for three DM strains. The major QTL, located on chromosome 2, consists of 12.95% of phenotypic variation explained (PVE) and a logarithm of odds (LOD) score of 14.12. Sixty-two candidate genes for *P. sorghi*, *P. maydis*, and *S. macrospora* resistance were obtained between the flanked markers in the QTL regions. The relative expression level of candidate genes was evaluated by quantitative real-time polymerase chain reaction (qRT-PCR) using resistant (CML228, Ki3, and Ki11) and susceptible (B73 and CML270) genotypes. For the 62 candidate genes, 15 genes were upregulated in resistant genotypes. Among these, three (*GRMZM2G028643*, *GRMZM2G128315*, and *GRMZM2G330907*) and *AC210003.2_FG004* were annotated as leucine-rich repeat (LRR) and peroxidase (POX) genes, respectively. These candidate genes in the QTL regions provide valuable information for further studies related to *P. sorghi*, *P. maydis*, and *S. macrospora* resistance.

## 1. Introduction

Maize is the world’s leading cereal in terms of production, with 1093 million metric tons produced on 186 million hectares globally. Maize is grown in both temperate and tropical areas of the world and is largely (around 80%) produced under rainy conditions in sub-Saharan Africa, South and Southeast Asia, and Latin America. It is particularly susceptible to abiotic and biotic stress. Eight major countries growing maize (China, India, Indonesia, Nepal, Pakistan, the Philippines, Thailand, and Vietnam) produce 98% of Asia’s and 28% of the world’s maize yield. Heavy economic and yield losses have been recorded due to infection by downy mildew (DM) agents in the Philippines, Taiwan, Indonesia, Thailand, India, Japan, Australia, Venezuela, North America, Europe, West Africa, and other parts of the world [1,2,3,4,5,6].

DM is caused by obligate pathogens that cannot be cultured in the laboratory, and its sporulation prefers high relative humidity, a night temperature of 21–23 °C, and light drizzle with cool weather. It is spread by oospores that survive in the soil [7] and can be spread through infected seeds or from plant-to-plant by airborne conidia. Because of the systemic nature of DM, susceptible lines usually die when infected in the seedling emergence stage, and when plants are infected during later growth stages, they cannot develop the maize ear despite having survived. At least six pathogens that cause DM infection of maize in Asia have been reported, including sorghum DM (*P. sorghi* (Weston & Uppal)), Philippine DM (*P. philippinensis* (Weston) Shaw), Java DM (*P. maydis* (Raciborski)), sugarcane DM (*P. sacchari* (Miyabe) Shirai and Hara), brown stripe DM (*Sclerophthora rayssiae* var. *zeae*), crazy top DM (*S. macrospora*), and Rajasthan DM (*P. heteropogoni*) [2,6,8,9,10,11,12,13,14,15]. DM is widespread in tropical regions, although its origin is conjectural, and because of the diversity of the DM pathogens and their systemic nature, the development of resistant varieties is needed. Moreover, a renewed emphasis on cost-effectiveness and environmental safety that has brought about the application of DM management by the development of resistant varieties. According to studies on the interaction of maize and the pathogens, resistance to DM is polygenically controlled [7,13,16,17,18,19,20,21,22,23].

Quantitative trait locus (QTL) mapping enables the detection, localization, and characterization of genetic factors contributing to polygenically inherited variation [24]. In QTL studies, the use of recombinant inbred lines (RILs) has more advantages than F_2_ or backcross populations [25,26,27]. Furthermore, RILs have been used to identify QTLs for the European corn borer [28], thermotolerance [29], and grain yield [30] in maize.

Several groups have performed QTL mapping using diverse mapping populations. George et al. [2] reported six QTLs on five chromosomes (1, 2, 6, 7, and 10) in RILs from the cross Ki3 (resistant) × CML139 (susceptible) of advanced inbred lines from the International Maize and Wheat Improvement Center (CIMMYT) in Indonesia, the Philippines, and Thailand. A QTL on chromosome 6 at bin 6.05 was found to majorly affect resistance to five DM pathogens (*P. sorghi, P. philippinensis, P. maydis, S. rayssiae* var. *zeae,* and *P. heteropogoni*). Agrama et al. [31] reported three QTLs on chromosomes 1 and 9 utilizing RILs derived from a cross between G62 (resistant) and G58 (susceptible) Egyptian inbreds. Two QTLs on chromosome 1 had a minor effect and one on chromosome 9 had a major effect. Sabry et al. [32] reported three QTLs on chromosomes 2, 3, and 9 utilizing F_3_ in Egypt, Thailand, and southern Texas. One QTL on chromosome 2 had a major effect and the two on chromosomes 3 and 9 had minor effects. Other studies have reported QTLs for *P. sorghi* (sorghum DM) resistance on chromosomes 2, 3, 4, 5, 6, and 9 [21,33,34].

Considering the complexity of quantitative traits, these QTLs can be used for introgression by marker-assisted selection with further validation [33]. It is difficult to estimate disease reaction accurately because of the factors influencing DM, such as plant maturity and the amount of pathogen inoculum. If there are differences in the pathogen populations or environment by genotype interactions in different locations, the analysis of simple and accurately scored molecular markers for the resistance genes of DM could greatly benefit future efforts to prevent loss to disease [32]. In addition, the evaluation of several DM strains using a mapping population could contribute to the accurate assessment of genetic contributions to resistance.

Genome-wide comparative transcriptome analysis has been performed by using an RNA-seq method in cabbage [35], cucumber [36], grapevine [37], and pearl millet [38]. DM resistance has been identified using the mutants at DM genes in lettuce [39]. Interaction between genes and pathways related to resistance against powdery mildew (PM) in melon has been profiled through comparative transcriptome analysis [40]. In maize, most studies on DM resistance have been performed using QTL analysis. In addition, there is a lack of information on the candidate genes and pathways for DM resistance in maize.

The main objective of this study is to validate candidate genes for *P. sorghi*, *P. maydis*, and *S. macrospora* resistance obtained from the QTL information of DM generated by using 192 F_7_ families derived from the cross B73 × Ki11. Our approach to the location of the QTLs for resistance to *P. sorghi*, *P. maydis*, and *S. macrospora* in maize is based on restriction fragment length polymorphism (RFLP) and simple sequence repeat (SSR) markers. We obtained candidate genes for *P. sorghi*, *P. maydis*, and *S. macrospora* resistance near the flanked markers of the QTL positions. The candidate genes were validated via quantitative real-time polymerase chain reaction (qRT-PCR) using resistant (CML228, Ki3, and Ki11) and susceptible (B73 and CML270) genotypes and identified by Pfam analysis.

## 2. Materials and Methods

### 2.1. Plant Materials

The mapping population consisting of RILs from the cross B73 (susceptible) and Ki11 (resistant) was evaluated in this study. Ki11 is the DM-resistant parent from the yellow flint line with late maturity. This inbred line is derived from Suwan1, which was developed for resistance to sorghum DM (*P. sorghi*) at Kasetsart University in Bangkok, Thailand [41,42]. B73 was the DM-susceptible parent from the temperate stiff-stalk maize line. This inbred line was developed at Iowa State University (Ames, IA) in the United States and is known to be susceptible to DM and drought, as well as being photoperiod-sensitive [43,44,45,46]. The F_1_ families were planted in 2013, and then F_7_ families of the RILs (*n* = 192) were obtained by selfing each individual plant in an experimental field at Dongguk University, Korea in 2017.

### 2.2. Evaluation of the RILs for DM

Field experiments with *P. sorghi* (sorghum DM)*, P. maydis* (Java DM)*, and S. macrospora* (crazy top DM) were conducted for genotypes (B73, CML228, CML270, Ki3, and Ki11) in Phnom Penh, Cambodia in April and September 2015. The sets of F_7_ RILs and parents (B73 and Ki11) were screened in in Phnom Penh, Cambodia in September to November 2017 (11°27′08.5″ N 105°09′57.6″ E; high humidity averages of 96.1%, 94.6%, and 93.4% and low humidity averages of 58.6%, 60.9%, and 63.1% in September, October, and November, respectively). DM disease was scored in the field using a modified spreader-row technique [2]. Susceptible genotypes (B73 and CML270) were used as two 5-m-long spreader rows with 20 plants per row after every 10th row of the test entries on the border rows of the experimental block (Appendix A) [44,47]. Seeds of the spreader genotypes germinated after 7 days, then a DM-infected maize plant in a pot was placed in each experimental block for 2 weeks to infect the spreader plants. After removing the DM-infected maize plant pots, RILs were planted with 50 × 25 cm spacing between the spreader rows. After germination of the RILs, DM incidence was assessed every 7 days for 6 weeks by scoring for systemic infection [48,49]:DM incidence (%) = (Number of infected plants/Total number of plants) × 100,(1)

The results are categorized as 0% (no symptoms) = highly resistant (HR), 1–10% = resistant (R), 11–25% = moderately resistant (MR), 26–50% = moderately susceptible (MS), 51–75% = susceptible (S), and 76–100% = highly susceptible (HS) [44].

Spreader genotypes were confirmed as having been 100% infected with DM, and the percentage of disease incidence was determined. The resistant plants did not show systemic symptoms of DM (emergence of characteristic chlorotic leaves) [50].

### 2.3. DNA Preparation

Genomic DNA was extracted from the parent plants and 192 F_7_ individuals at the four-leaf stage using a modified DNA isolation protocol. The quality and quantity of the genomic DNA were analyzed with 1% agarose gel electrophoresis and Spectrophotometer (Model MN-913, Maestrogen, Hsinchu, Taiwan).

### 2.4. Molecular Marker Assay

A total of 727 SSR and RFLP markers covering all 10 chromosomes were obtained from the Maize Genetics and Genomics Database (MaizeGDB) and publications. The amplification was performed using a Takara PCR Thermal Cycler Dice Touch (Takara, Shiga, Japan) with 25 ng of gDNA and Takara Taq polymerase (Takara, Shiga, Japan). The polymerase chain reaction (PCR) conditions consisted of one cycle of 5 min at 94 °C for the initial denaturation and 30 cycles of 1 min at 94 °C, 1 min at 51–65 °C (based on the annealing temperatures standardized for different primers), and 30 s at 72 °C, with a final extension step for 10 min at 72 °C. The PCR products were separated by electrophoresis on 3% agarose gel with 1× Tris-acetate-ethylenediaminetetraacetic acid buffer at 100 V for 1 h and visualized using ethidium bromide staining. Markers were screened for polymorphism between the parent lines of the RILs. Of the 727 markers, 234 were found to be polymorphic based on agarose gel electrophoresis; these were used for genotyping the 192 RIL individuals (Appendix A). Alleles of the parents of B73 and Ki11 were designated as 2 and 0, respectively.

### 2.5. Linkage Map Construction and QTL Analysis for P. sorghi, P. maydis, and S. macrospora Resistance

QTL IciMapping v. 4.1 software from the Quantitative Genetics Group was used for both linkage map construction and QTL analysis using the MAP and BIP functions, respectively [51,52]. For linkage map construction, groups were ordered with an LOD score of 3.0 and the nearest neighbor combined with the two-opt algorithm (nnTwoOpt). Rippling was ordered with the sum of adjacent recombination frequencies with a window size of 5. Recombination frequencies were transformed into cM distances between linked loci using Kosambi’s [53] mapping function. For QTL analysis, inclusive composite interval mapping of the additive (ICIM-ADD) function was applied with 1.0 step and 0.001 probability in the stepwise regression. The data of missing phenotypes were ordered with the Deletion command. The threshold LOD scores were calculated using 1,000 permutations with a type I error of 0.05 [51,54]. Suggestive QTLs with an average LOD value >3.0 in a dataset were noted. A QTL with an average LOD value > 3.0 and average phenotypic variance contribution > 10% was defined as a major QTL [55,56]. The QTLs in the present study were compared with previously published ones [2,21,31,32]. This software program uses an improved algorithm of composite interval mapping with increased power to detect QTLs to reduce false detection rates and create less biased QTL effect estimates by employing stepwise regression followed by QTL scanning [51].

### 2.6. RNA Extraction and Candidate Gene Screening with Quantitative Real-Time PCR (qRT-PCR)

Total RNA of healthy (control) and DM-infected parent lines was extracted from 6-week-old leaves of B73, CML228, CML270, Ki3, and Ki11 using TRIzol (Invitrogen, Carlsbad, CA, USA). For the qRT-PCR, cDNA was synthesized using ReverTra Ace^®^ qPCR RT Master Mix with gDNA Remover (Toyobo, Osaka, Japan) using 1 μg of total RNA extracted from leaves.

We searched for annotated genes near 14 flanked markers in the reference genome of B73 RefGen_v2 (MaizeGDB 2019). Partially, annotated genes were reported to be associated with biotic and abiotic stress. We obtained 62 genes in seven QTL locations; these were spread over chromosomes 2, 3, 6, and 9. The information and predicted protein of candidate genes were obtained from maizeGDB (B73 RefGen_v2). In addition, 19 DM-related genes from *Arabidopsis thaliana*, *Oryza sativa*, and *Zea mays* were obtained from the NCBI database (2019). The qRT-PCR primer sets were designed based on the 62 candidate genes and 19 DM-related genes using Primer3 [57] to validate the expression value of each candidate genes compared to control and DM-infected maize (Appendix A). The qRT-PCR was performed by using a gene-specific primer set, the CFX96 Touch Real-Time PCR Detection System (Bio-Rad Laboratories, USA). The qRT-PCR mixture contained 20 ng of cDNA, 0.2 µM of each gene-specific primer, 8 µL of sterile water, and 10 µL of TOPreal ™ qPCR 2× PreMIX (containing SYBR Green with low ROX; Enzynomics, South Korea) in a total volume of 20 μL. The qRT-PCR conditions were 10 min at 95 °C for the initial denaturation, then 45 cycles of 10 s at 95 °C, 10 s at 55–60 °C (based on the annealing temperatures standardized for different primers), and 30 s at 72 °C. The maize 18s rRNA (*AF168884.1*), *UBCP* (*GRMZM2G102471*_T01), *MEP* (*GRMZM2G018103*_T01), and *LUG* (*GRMZM2G425377*_T01) were used as reference genes (Appendix A) [58]. The specificity and efficiency of the amplicon were confirmed through the melting curve analysis from 65 to 95 °C after each qRT-PCR. Three replicates of each experiment were performed for each candidate gene. The relative gene expression compared with the control plant was calculated using the 2^−ΔΔ*C*t^ method [59].

## 3. Results

### 3.1. Phenotypic Data for the DM Analysis

In this study, the DM incidence of DM-resistant genotypes (CML228, Ki3, and Ki11) and DM-susceptible genotypes (B73 and CML270) are reported as phenotypically similar between April and September 2015 (Table 1). Both B73 and CML270 showed 100% DM disease incidence within 6 weeks after inoculation in April and September, while comparable figures for CML228, Ki3, and Ki11 were 0%/25%, 0%/22.2%, and 5%/25%, respectively. According to the weather records, September was wetter and cooler than April.

We evaluated resistance to *P. sorghi, P. maydis,* and *S. macrospora* using 192 F_7_ families derived from B73 (susceptible) × Ki11 (resistant) since we decided that the approach of polygenic control for the various DM strains would offer more effective pathogen resistance (Appendix A).

We performed DM screening of F_7_ families in September 2017 because DM development is favorable under high humidity and low temperature conditions [60]. The mean of the DM incidence for the F_7_ families ranged from 0% to 100%, and the distribution of the F_7_ families was skewed toward the susceptible lines (a RILs mean of 86.49% for DM incidence) (Figure 1 and Table 2).

### 3.2. Marker Data Analysis and Linkage Mapping

We screened 691 SSR and 36 RFLP markers to identify polymorphisms between B73 and Ki11. A total of 228 SSR and 6 RFLP markers were used to construct a linkage map of F_7_ families. The map covered around 2042.51 cM at an average marker interval of 9.12 cM and 79 bins among the approximate 120 bin locations on 10 chromosomes. Around 85.04% of the markers were within 20 cM of the nearest interval. Chromosome 2 had the highest number of markers (34), while chromosome 7 had the lowest (14) (Appendix A).

### 3.3. QTL Analysis

QTLs were classified using logarithm of odds (LOD) scores exceeding the 3.0 threshold. The seven QTLs for *P. sorghi*, *P. maydis*, and *S. macrospora* resistance were detected by constructing a linkage map and analyzing the LOD scores for the F_7_ families (B73 × Ki11) by ICIM-ADD using QTL IciMapping (Figure 2, Appendix A, and Table 3). The seven QTLs were located on chromosomes 2 (bins 2.01 and 2.02), 3 (bins 3.04 and 3.05), 6 (bin 6.05/6.06), and 9 (bins 9.05 and 9.07). All of the QTLs were contributed by the resistant parent (Ki11). The LOD scores and phenotypic variation explained (PVE) values of the QTLs ranged from 3.17 to 18.16 and from 0.47% to 12.95%, respectively. The QTL detected on bin 2.01 had a major effect whereas the ones on bins 2.02, 3.04, 3.05, 6.05/6.06, 9.05, and 9.07 contributed minor effects (the QTLs were classified with PVE values higher than 10.0% as major and less than 10.0% as minor) [49]. The major QTL on *qDM1* (2.01) flanked by umc1165 and bnlg1297 presented 12.95% of the phenotypic variation for *P. sorghi*, *P. maydis*, and *S. macrospora* resistance. The other minor QTLs on *qDM2* (2.02), *qDM3* (3.04), *qDM4* (3.05), *qDM5* (6.05/6.06), *qDM6* (9.05), and *qDM7* (9.07) flanked by umc2363-phi098, umc1030-phi243966, mmc0022-bnlg420, bnlg345-umc1859, umc1231-umc2343, and dupssr29-umc1505 presented 0.75%, 0.48%, 2.90%, 2.85%, 0.77%, and 0.47% of the phenotypic variation, respectively.

### 3.4. qRT-PCR Validation of the Candidate Genes for P. sorghi, P. maydis, and S. macrospora Resistance

We performed an analysis of the expression levels of reference genes (18s rRNA genes, *UBCP*, *MEP*, and *LUG*) [58]. Nineteen DM-related genes from *Arabidopsis thaliana*, *Oryza sativa*, and *Zea mays* (*LOC4345959*, *LOC107275863*, *SGT1B*, *LOC103650325*, *LOC100382073*, *LOC103647182*, *LOC4351808*, *HSK*, *LOC4324025*, *LOC103654479*, *LOC4345309*, *DMR6*, *IDC1*, *LOC103632498*, *LOC100191339*, *LOC107275878*, *LOC103648264*, *LOC103642860*, and *EDM2*) were validated by comparing control and DM-infected plants that were DM-resistant (CML228, Ki3, and Ki11) or DM-susceptible (B73 and CML270) genotypes (Table 4). Five genes (*LOC107275878*, *LOC103632498*, *LOC103647182*, *HSK*, and *LOC4345309*) were upregulated by DM in the DM-resistant genotypes, while another five genes (*LOC103648264*, *LOC4324025*, *LOC100191339*, *LOC100382073*, and *IDC1*) were partially upregulated (Figure 3). The other nine genes were not expressed in the control or DM-infected genotypes. We considered that gene expression levels showed a wide range of variation because resistant genotypes (CML228, Ki3, and Ki11) differ in origin and pedigree.

The candidate genes related to *P. sorghi*, *P. maydis*, and *S. macrospora* resistance were obtained by searching for all of the flanked markers in the QTL regions in MaizeGDB (B73 RefGen_v2) (Table 5 and Appendix A). The genes were flanked by 14 markers (umc1165, bnlg1297, umc2363, phi098, umc1030, phi243966, mmc0022, bnlg420, bnlg345, umc1859, umc1231, umc2343, dupssr29, and umc1505). Three markers (bnlg1297, umc2363, and phi098) were identified by the same physical location in B73 RefGen_v2, and so we suggest that *qDM1* and *qDM2* are likely the same. *qDM1* was analyzed as a major QTL, thus we supposed that DM-related genes were possible to locate in *qDM1* region. *qDM1* and *qDM2* are located at 4,051,998–4,407,701; 37 genes were obtained in the region. Of the 37 genes, 7 were identified as short sequence length or without genetic information. The relative expression levels of the 62 genes by qRT-PCR were validated by comparing the control and DM-infected plants from B73, CML228, CML270, Ki3, and Ki11. However, 17 genes were not expressed in the control or DM-infected genotypes.

From the 45 genes, 15 (*GRMZM2G128315, AC210003.2_FG004, GRMZM2G045049, GRMZM2G178880, GRMZM2G363066, GRMZM2G028643, GRMZM2G330907, GRMZM2G047677, AC191071.3_FG001, GRMZM2G133707, GRMZM2G020043, GRMZM2G039345, GRMZM2G314171, GRMZM2G062031,* and *GRMZM2G005984*) were upregulated by DM in the DM-resistant genotypes (Figure 4 and Appendix A). These 15 upregulated genes were classified as peroxidase (POX), chloroplastic/mitochondrial phosphoribosylformylglycinamidine (PITG) synthase, mannan synthase (ManS), G-type lectin S-receptor-like serine/threonine protein kinase (GsSTK), LRR receptor-like STK (LRR-STK), putative LRR receptor-like protein kinase (LRR-RLK) family protein, LRR family protein, putative STRUBBELIG family receptor protein kinase, abscisic acid receptor PYL5, abscisic acid receptor PYL3, pyrabactin resistance-like protein (PYL), probable flavin-containing monooxygenase 1 (FMOs), P-loop containing nucleoside triphosphate hydrolase superfamily protein (NTPase), photosystem II repair protein PSB27-H1 chloroplastic, photosystem II protein, and uncharacterized. The uncharacterized genes (*GRMZM2G062031, GRMZM2G133707, GRMZM2G20043,* and *GRMZM2G314171*) were highly expressed in DM-infected plants. Of the 45 genes, 30 were partially upregulated and were classified into 20 functional annotations.

## 4. Discussion

DM, caused by *Peronosclerospora* species in maize, causes severe yield loss despite the use of chemical pesticides and has spread in many tropical and subtropical regions throughout the world. The lack of suitable studies for gene diversity in DM has been a major constraint in tropical Asia, especially in the maize-growing environments of South and Southeast Asia [61]. The results from a great many studies on *P. sorghi* (sorghum DM) resistance have been reported because it is widely distributed throughout Asia, Africa, and America, where it is easier to perform experiments than in other regions. However, we need to evaluate various DM species because the fungus can easily spread to other regions through the movement of spores in the air.

We screened resistance for *P. sorghi* (sorghum DM)*, P. maydis* (Java DM)*,* and *S. macrospora* (crazy top DM) using 192 F_7_ families derived from B73 (susceptible) × Ki11 (resistant) in Phnom Penh, Cambodia. The distribution of the F_7_ families was skewed toward the susceptible lines. According to previous studies, it is not uncommon that phenotype values of mapping populations do not follow a normal distribution [2,21,30]. The phenotype data for the DM of the F_7_ families leaned more toward susceptible because of the response to the three DM pathogens. We used 691 SSR and 36 RFLP markers from MaizeGDB (B73 RefGen_v2) for parental polymorphism. We performed polymorphism analysis by electrophoresis using 3% agarose gel with 1× TAE buffer at 100 V for 1 h. It is necessary that a difference of greater than 20 bp between PCR products could be detected by using 3% agarose gel and loading for 1 h. Additionally, we analyzed flanked markers in the QTL regions by using QiaXcel advanced system (Qiagen, Hilden, Germany). The constructed linkage map covered around 2042.51 cM at average intervals of 9.12 cM between markers for 228 SSR and six RFLP markers on 10 chromosomes. Around one-third of the markers showed to be polymorphic; this result was considered an average ratio when compared to references [21,32,33,34,62]. These markers were distributed over 79 bins among around 120 bin locations. Approximately 85.04% of the markers were within 20 cM of the nearest interval (Appendix A). In QTL analysis, it is likely that RILs (F_7_ families (B73 × Ki11)) made a significant contribution because the use of RILs is more powerful than F_2_ or backcross populations in QTL analysis [25,26,27]. However, in this study, phenotype data for *P. sorghi*, *P. maydis*, and *S. macrospora* resistance showed recessive trait. In analysis of recessive traits, backcross inbred lines (BILs) derived from resistant genotype have more power to detect recessive QTLs because they have an advantage in segregation ratio compared to F_2_ or RILs [63].

The seven QTLs for *P. sorghi*, *P. maydis*, and *S. macrospora* resistance were identified on 4 of 10 chromosomes (Table 3). A major QTL was identified in bin 2.01 (*qDM1*) on chromosome 2, while the other QTLs were detected in bins *qDM2* (2.02), *qDM3* (3.04), *qDM4* (3.05), *qDM5* (6.05/6.06), *qDM6* (9.05), and *qDM7* (9.07). All of the DM-resistant alleles were obtained from Ki11. According to previous studies, the QTLs for DM resistance have been detected in chromosomes 2, 3, 6, and 9, and several flanked markers of QTL regions have been reported. We considered that these seven QTLs affected *P. sorghi* (sorghum DM), *P. maydis* (Java DM), and *S. macrospora* (crazy top DM) from Cambodia. The QTL regions found in the present study were similar to those in previous studies for *P. sorghi* (sorghum DM) and *P. heteropogoni* (Rajasthan DM) resistance [21,32,33,34,49] (Appendix A). *qDM1* and *qDM2* shared the same flanked markers (umc1165 and umc2363 in bins 2.01 and 2.02, respectively) [34]. *qDM3* had flanked marker umc1030 in bin 3.04 [62]. *qDM4* and *qDM5* shared the same flanked markers (bnlg420 and umc1859 in bins 3.05 and 6.05/6.06, respectively) [21]. In addition, *qDM6* and *qDM7* shared the same flanked markers (umc2343 and dupssr29 in bins 9.05 and 9.07, respectively) [33]. The bin locations (3.04 and 3.05, respectively) of *qDM3* and *qDM4* matched with other reports on QTLs [21,32,33,49]. The other QTLs were identified from five different DM strains (*P. sorghi* and *P. heteropogoni* from India, *P. zeae* from Thailand, *P. philippinensis* from the Philippines, and *P. maydis* from Indonesia) and sorghum DM on bin 6.05 using RILs (derived from Ki3) and backcross populations [2,21]. Ki3 and Ki11 were developed as DM-resistant lines derived from the Suwan1 strain from Thailand. These results suggest that it should be possible to detect candidate genes for *P. sorghi*, *P. maydis*, and *S. macrospora* resistance near these bin positions. Although the QTL analysis was performed using different mapping populations, the results of this study are well matched with previously reported ones.

We analyzed 19 DM-related genes from *Arabidopsis thaliana*, *Oryza sativa*, and *Zea mays* (Table 4): *EDM2*, *SGT1B*, *LOC100191339*, *HSK*, *DMR6*, *IDC1*, and *LOC103642860*, among others. Among the 19 genes, five were upregulated in the DM-resistant genotypes, two of which (*LOC103632498* and *LOC103647182*) are located on chromosomes 2 and 7 of maize, respectively. *LOC100191339* was located on nearby *qDM4* and was highly upregulated in DM-infected CML270 (susceptible). Two genes (*LOC107275878* and *LOC4345309*) and *HSK*, originating from *Arabidopsis thaliana* and *Oryza sativa*, were significantly upregulated in the DM-infected DM-resistant genotypes. Hence, we considered that these genes are conserved in monocotyledon and dicotyledon plants [64,65].

We obtained 62 candidate genes for *P. sorghi*, *P. maydis*, and *S. macrospora* resistance near the flanked markers in the QTL region; these genes were validated by comparing their relative expression levels in the control and DM-infected groups. The physical locations of candidate genes were continually updated from B73 RefGen_v1 to B73 RefGen_v4, but updating the genetic information of the genomic markers is slower than the transcripts. Hence, we used B73 RefGen_v2 to set the physical locations of the genomic markers and candidate genes. The annotations of 45 genes were analyzed to predict their functions using the Pfam database. The annotations and functions of these genes were identified as being related to DM resistance (Table 6 and Appendix A). There are various factors involved in DM resistance, such as POX (peroxidase), MYB (transcriptional activator Myb), GSO1 (LRR receptor-like serine/threonine protein kinases (STKs), (LRR-STKs)), plant RLKs, polygalacturonase inhibitor protein (PGIP), polyphenol oxidase (PPO), NAC, WRKY transcription factors, and pathogenesis-related (PR) protein [66,67,68,69,70,71,72].

Three genes (*AC210003.2_FG004*, *AC191071.3_FG001*, and *GRMZM2G039345*) were annotated as POX, FMO, and RuBisCO (ribulose-1,5-bisphosphate carboxylase/oxygenase), respectively. The activities of POX and PPO, along with b-1,3-glucanase, are associated with DM resistance in sunflowers [73]. Plants can implement DM post-infection mechanisms such as an increase in localized callose deposition to fortify plant cell walls [74,75], reactive oxygen species (ROS), peroxidase activity, and hypersensitive response activation [74,76]. When a resistant grapevine is infected with DM, it produces high concentrations of resveratrol that can be oxidized by induced peroxidase [77]. In broccoli, BoAPX (ascorbate peroxidase) genes contribute enhanced both DM and heat tolerance, and play important roles in cellular defense against ROS-mediate oxidative damage [68]. In previous studies, high POX activity has been associated with resistance to PM [78] in lettuce [79] and melon [80] and to *Verticillium dahlia* in tomatoes [81]. PM is very similar to DM in that both are caused by fungi and are widespread under humid and low temperature conditions; the disease symptoms are also similar. FMO1 positively regulates the enhanced disease susceptibility1 (EDS1) pathway in *Arabidopsis thaliana*. The EDS1 pathway controls defense activation and programmed cell death against pathogens [82]. In addition, a defect in FMO1 partially disables toll interleukin 1 receptor nucleotide binding sites leucine-rich repeat (TIR-NB-LRR) resistance and basal defense. In *Arabidopsis thaliana*, RPP4, which has been identified as a TIR-NB-LRR protein coupled with its dependence on signaling components in leaves, confers resistance to DM [83]. RPP4-mediated resistance is regulated by interaction between EDS1 and NDR1 signaling in cotyledons.

Two genes (*GRMZM2G342564* and *GRMZM2G040095*) were identified as lipoxygenase (LOX)-producing. LOX is known to play a role in disease resistance for many host pathosystems. In earlier reports, LOX activity was found to increase in resistant plants and to decrease in susceptible plants. Also, LOX was reported to affect DM and PM resistance of pearl millet and wheat, respectively [84,85,86,87,88].

*GRMZM2G028643*, *GRMZM2G128315*, and *GRMZM2G330907* (identified as LRR) and *GRMZM2G363066* (identified as nonspecific STK) were associated with defense reactions against pathogens. PR protein are encoded by disease resistance (R) genes, responding to pathogenic microorganisms and signaling cascades that activate defense reactions [89,90]. The largest family of PR proteins is defined by the presence of 12 to 21 LRRs (it has been speculated that LRRs bind pathogen-derived ligands). A glutamate-to-lysine substitution in LRR partially compromises the function of R genes against DM [91]. Several kinases (protein kinases, wall-associated kinases, calcium-dependent protein kinases, STKs, LRR-STKs, and mitogen-activated protein kinases (MAPKs)) are strongly associated with signal transduction mediated by Ca^2+^ permeable channels [90,92,93,94]. It has been shown that the activation of the PR protein and phenylpropanoid pathway enzymes such as LRR-STKs and MAPKs responds to DM infection in pearl millet by inducing molecules for signal transduction [95]. Also, nucleotide-binding site (NBS)-LRR and receptor-like proteins (RLP) were included in five classes of *R* genes, these have been reported as resistance against DM [70,71]. RLKs are well known to play a role in many important signaling process such as plant growth, development, hormone signaling, and stress response. LRR-RLK family proteins regulate plant innate immunity and defense [69]. PGIP is a defense protein consisting of an extra-cytoplasmic LRR which specifically binds the invading fungus cell wall to the host tissue [96]. In previous studies, the transcription of PGIP and a cell wall glycoprotein were induced in DM-resistant strains of pearl millet [38] and grape [97]. In *Arabidopsis thaliana*, homoserine accumulation in the chloroplast triggers a novel form of DM resistance that is independent of known immune responses [98]. In pearl millet, the role of PGIP in resistance against DM pathogen (*Sclerospora graminicola*) was reported by differential gene expression analysis between resistant and susceptible genotypes [72].

*GRMZM2G005984* was classified as a photosystem II protein. In grapevine, PM-responsive proteins are involved in photosynthesis, metabolism, disease/defense, protein destination, and protein synthesis [99]. These proteins are associated with the plant defense response and slow down disease progression against *Erysiphe necator*. *GRMZM2G314171* was classified as DEAD-box RNA helicase, one of which (OsBIRH1) has been found to modulate the defensive response to infection and oxidative stress in rice [100]. The DEAD-box RNA helicase family functions in chloroplast biogenesis in maize [101]. *GRMZM2G178880* was classified as a GTF (general transcription factor); these proteins are probably involved in defense-related processes [102]. It has been shown that GTF expression increases during the interaction between PM and barley [103], and its expression level is induced after wounding and phytopathogenic bacteria attack [104]. *GRMZM2G024293* was identified as a conserved hypothetical adenosine triphosphate (ATP)-binding protein. Its downregulation brings about a decrease in ATP binding in DM-infected plants, which makes it possible to estimate DM susceptibility. Meanwhile, further study is needed to identify the functional characteristics of the uncharacterized genes.

Hence, we suggest that 10 genes (*AC210003.2_FG004, AC191071.3_FG001, GRMZM2G039345, GRMZM2G028643, GRMZM2G128315, GRMZM2G330907, GRMZM2G363066, GRMZM2G005984, GRMZM2G178880, and GRMZM2G314171*) are related to *P. sorghi*, *P. maydis*, and *S. macrospora* resistance. The downregulation of some of the *P. sorghi*, *P. maydis*, and *S. macrospora* resistance genes occurs in DM-infected plants. Moreover, it was shown that 30 genes are partially upregulated in DM-infected plants of resistant genotype. From these results, we predict the possibility that the phenotype of resistant plants shows resistance (R) or moderate resistance (MR).

Experiments on *P. sorghi, P. maydis, and S. macrospora* resistance have been limited to QTL analysis in previous studies. However, we focused on their great importance through the screening of candidate genes using qRT-PCR. The results of this study can serve as fine-mapping for DM and marker-assisted selection (MAS) in maize breeding. In further research, our approach could be used to screen other DM strains in different environments using RILs and to validate candidate genes using CRISPR/Cas9. In addition, the genes and pathways associated with resistance could be identified via comparative transcriptome profiling using RNA-seq.

## Figures and Tables

**Figure 1 genes-11-00191-f001:**
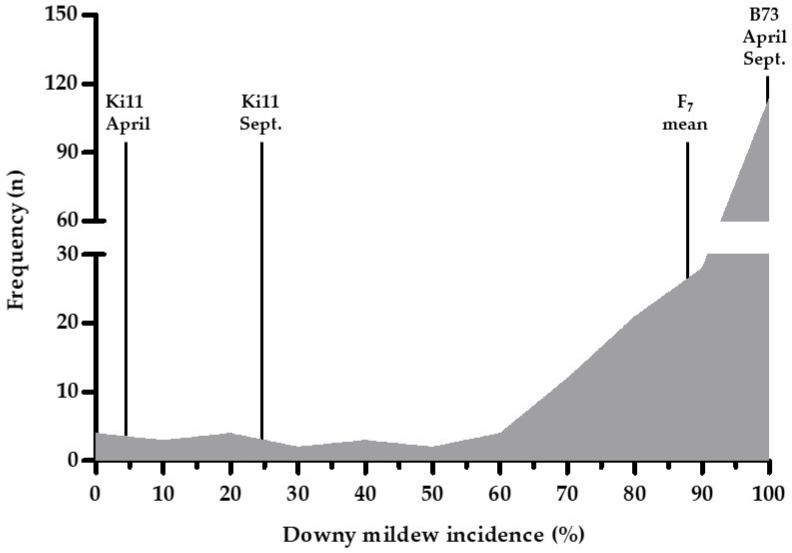
Frequency distribution of the F_7_ families derived from B73 (susceptible) × Ki11 (resistant) based on downy mildew (DM) incidence in Phnom Penh, Cambodia in September 2017. The *x*-axis indicates the frequency of DM incidence in F_7_ families. Ki11 and B73 were evaluated in April and September 2015.

**Figure 2 genes-11-00191-f002:**
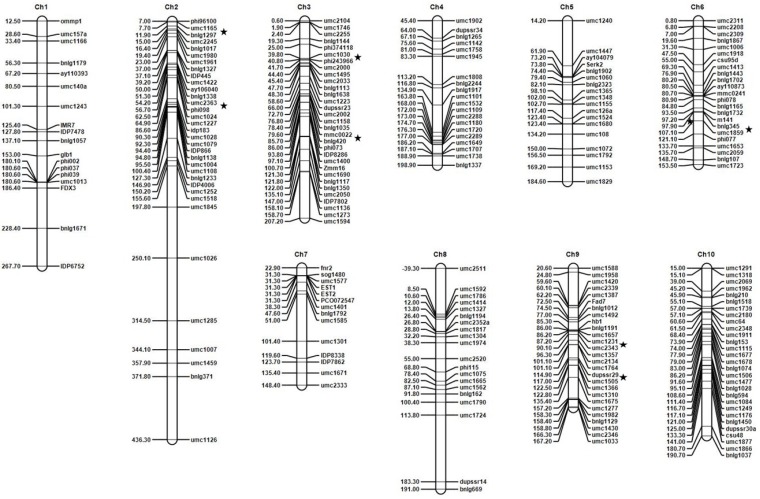
The linkage map of the F_7_ families of the cross between B73 (susceptible) and Ki11 (resistant). For each chromosome, the chromosome number is shown at the top, the markers on the right side, and the genetic distances in cM calculated using the Kosambi function on the left side. ★, a quantitative trait locus (QTL) position.

**Figure 3 genes-11-00191-f003:**
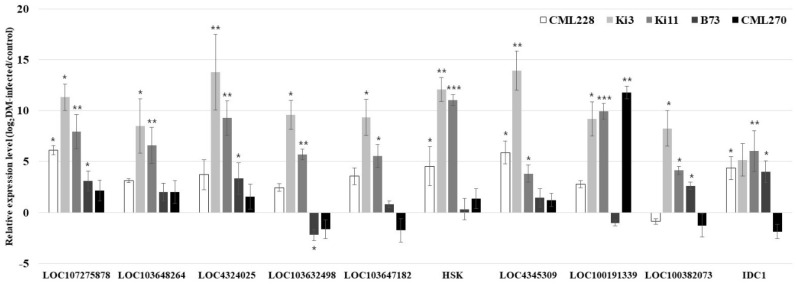
Relative expression levels (log_2_DM-infected/control) of 10 *P. sorghi*, *P. maydis*, and *S. macrospora* related genes in the control and infected plants measure via quantitative real-time polymerase chain reaction (qRT-PCR). The analysis was conducted on three independent plants as biological replicates. The data are presented as the mean ± standard error (SE). Student’s *t*-tests were used in the statistical analysis (* *p* < 0.05; ** *p* < 0.01; *** *p* < 0.001).

**Figure 4 genes-11-00191-f004:**
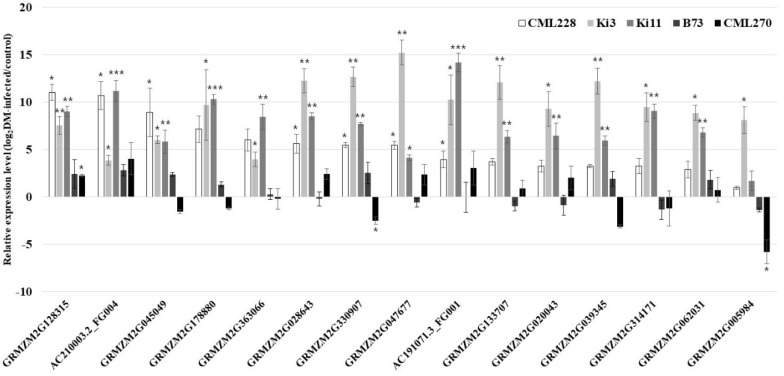
Relative expression levels (log_2_DM-infected/control) of 15 candidate genes for *P. sorghi*, *P. maydis*, and *S. macrospora* via qRT-PCR in the control and infected plants. The analysis was conducted on three independent plants as biological replicates. The data are presented as the mean ± SE values. Student’s *t*-tests were carried out in the statistical analysis (* *p* < 0.05; ** *p* < 0.01; *** *p* < 0.001).

**Table 1 genes-11-00191-t001:** Downy mildew (DM) Incidence of resistant (CML228, Ki3, and Ki11) and susceptible (B73, and CML270) parents in Phnom Penh, Cambodia 2015.

Variety	Mid-April to June		Early September to October	
	4 Weeks	6 Weeks	4 Weeks	6 Weeks
**Resistant Genotype**				
CML228	0%	0%	20%	25%
Ki3	0%	0%	16.6%	22.2%
Ki11	0%	5%	10%	25%
**Susceptible Genotype**				
B73	50%	100%	100%	100%
CML270	100%	100%	75%	100%

0%, highly resistant (HR); 1–10%, resistant (R); 11–25%, moderately resistant (MR); 26–50%, moderately susceptible (MS); 51–75%, susceptible (S); 76–100%, highly susceptible (HS).

**Table 2 genes-11-00191-t002:** Statistical data for DM experiments conducted in Phnom Penh, Cambodia in September 2017.

Trait	Sample Size	Mean of DM Incidence	Standard Deviation	Minimum	Maximum	Range	Skewness	Kurtosis
DM	192	86.49	23.87	0	100	0–100	−2.35	5.14

**Table 3 genes-11-00191-t003:** Quantitative trait loci (QTLs) for *P. sorghi*, *P. maydis*, and *S. macrospora* resistance for the F_7_ families derived from B73 (susceptible) × Ki11 (resistant).

QTLs	Chr ^1^	Bin Location	Position	Left CI	Right CI	Left Marker	Right Marker	LOD ^2^	PVE (%) ^3^	Add ^4^	Donor of DM-Resistant Genotype
*qDM1*	2	2.01	11.03	10.53	11.53	umc1165	bnlg1297	14.12	12.95	−32.87	Ki11
*qDM2*		2.02	55.03	54.53	55.53	umc2363	phi098	3.60	0.75	−13.93	Ki11
*qDM3*	3	3.04	40.60	40.10	41.10	umc1030	phi243966	3.21	0.48	−8.39	Ki11
*qDM4*		3.05	80.60	80.10	81.10	mmc0022	bnlg420	18.16	2.90	−31.68	Ki11
*qDM5*	6	6.05/6.06	98.80	98.30	99.30	umc1859	bnlg345	10.77	2.85	−33.18	Ki11
*qDM6*	9	9.05	88.58	88.08	90.08	umc1231	umc2343	3.84	0.77	−11.09	Ki11
*qDM7*		9.07	116.58	115.08	117.08	dupssr29	umc1505	3.17	0.47	−8.27	Ki11

^1^ Chr: chromosome; ^2^ LOD: logarithm of odds; ^3^ PVE: phenotype variance explained; ^4^ Add: additive effects by the alleles of Ki11 and B73.

**Table 4 genes-11-00191-t004:** List of the DM-related genes in *Arabidopsis thaliana*, *Oryza sativa*, and *Zea mays* obtained from the NCBI Database (2019).

Gene ID	Cultivar	Chromosome	Location	Description
EDM2	*A. thaliana*	5	22,447,937–22,454,805	Enhanced downy mildew 2
SGT1B	*A. thaliana*	4	6,851,184–6,853,912	SGT1b
HSK	*A. thaliana*	2	7,508,473–7,509,887	Homoserine kinase; downy mildew resistance 1
DMR6	*A. thaliana*	5	8,378,759–8,383,401	Putative 2OG-Fe(II) oxygenase
LOC4324025	*O. sativa*	1	17,932,375–17,947,080	Enhanced downy mildew 2
LOC107275878	*O. sativa*	3	8,811,797–8,818,014	Enhanced downy mildew 2
LOC107275863	*O. sativa*	8	15,057,975–15,070,064	Enhanced downy mildew 2-like
LOC4345309	*O. sativa*	8	15,114,902–15,130,493	Enhanced downy mildew 2
LOC4345959	*O. sativa*	8	24,800,574–24,812,015	Enhanced downy mildew 2
LOC4351808	*O. sativa*	12	6,760,146–6,763,038	Enhanced downy mildew 2
LOC103647182	*Z. mays*	2	160,527,731–160,580,975	Enhanced downy mildew 2
LOC103648264	*Z. mays*	2	241,693,442–241,703,192	Enhanced downy mildew 2
LOC100191339	*Z. mays*	3	135,359,547–135,361,124	Downy mildew resistance 6
LOC103650325	*Z. mays*	3	102,996,061–102,998,690	Enhanced downy mildew 2
LOC100382073	*Z. mays*	4	173,897,007–173,895,527	Downy mildew resistance 6
LOC103654479	*Z. mays*	4	202,734,383–202,764,701	Enhanced downy mildew 2
IDC1	*Z. mays*	6	20,483,352–20,484,802	Iron deficiency candidate 1; downy mildew resistance 6
LOC103632498	*Z. mays*	7	79,090,176–79,133,359	Enhanced downy mildew 2
LOC103642860	*Z. mays*	10	135,816,580–135,824,835	Flavanone 3-dioxygenase 2

**Table 5 genes-11-00191-t005:** List of the *P. sorghi, P. maydis, and S. macrospora* resistance candidate genes between the left and right flanked markers.

Transcript ID	Chr ^1^	Bin Location	Type	Length (bp)	Predicted Protein Size (aa)	Description	Protein Information
AC210003.2_FG004	2	2.01/2.02	T01	999	332	Uncharacterized LOC100274427	Peroxidase 16
GRMZM2G020043			T01	1938	260	-	-
GRMZM2G039345			T01	697	206	Ribulose bisphosphate carboxylase/oxygenase activase 2, chloroplastic	P-loop containing nucleoside triphosphate hydrolase superfamily protein
GRMZM2G045049			T01	2111	521	Probable phosphoribosylformylglycinamidine synthase, chloroplastic/mitochondrial	Probable phosphoribosylformylglycinamidine synthase, chloroplastic/mitochondrial
GRMZM2G314171			T01	1395	464	-	-
GRMZM2G363066			T01	1450	422	G-type lectin S-receptor-like serine/threonine-protein kinase At1g34300	G-type lectin S-receptor-like serine/threonine-protein kinase At1g34300
GRMZM2G133707	3	3.04	T01	217	-	-	-
GRMZM2G047677	6	6.05/6.06	T01	1199	271	Uncharacterized LOC100216590	Abscisic acid receptor PYL5; abscisic acid receptor PYL3; pyrabactin resistance-like protein
GRMZM2G062031			T01	1809	382	Uncharacterized LOC100276496	Uncharacterized LOC100276496
GRMZM2G128315			T01	3343	964	Probable LRR receptor-like serine/threonine-protein kinase IRK	Putative leucine-rich repeat receptor-like protein kinase family protein
GRMZM2G005984	9	9.05	T01	951	222	Photosystem II 11 kd protein	Photosystem II repair protein PSB27-H1 chloroplastic; photosystem II protein
AC191071.3_FG001		9.07	T01	1605	534	Probable flavin-containing monooxygenase 1	Probable flavin-containing monooxygenase 1
GRMZM2G028643			T01	1954	523	Putative leucine-rich repeat receptor-like serine/threonine-protein kinase At2g14440	Putative leucine-rich repeat receptor-like serine/threonine-protein kinase At2g14440; Leucine-rich repeat (LRR) family protein
GRMZM2G178880			T01	1894	574	Uncharacterized LOC100191890	Putative mannan synthase 7
GRMZM2G330907			T01	3005	759	Uncharacterized LOC541659	Leucine-rich repeat transmembrane protein kinase 3; leucine-rich transmembrane protein kinase2; putative STRUBBELIG family receptor protein kinase

^1^ Chr: chromosome.

**Table 6 genes-11-00191-t006:** Pfam (EBI 2019) Domain Analysis of the 15 Upregulated Genes via qRT-PCR.

Transcript ID	Family	Description
AC210003.2_FG004	Peroxidase	Peroxidase (PF00141)
GRMZM2G020043	Hydrolase	Haloacid dehalogenase-like hydrolase (PF00702)
GRMZM2G039345	-	-
GRMZM2G045049	AIRS_C, GATase_5	AIR synthase related protein, C-terminal domain (PF13507); CobB/CobQ-like glutamine amidotransferase domain (PF13507)
GRMZM2G314171	DEAD, Metalloenzyme, Metalloenzyme	DEAD/DEAH box helicase (PF00270); Metalloenzyme superfamily (PF01676); Metalloenzyme superfamily (PF01676)
GRMZM2G363066	Pkinase_Tyr	Protein tyrosine kinase (PF07714)
GRMZM2G133707	-	-
GRMZM2G047677	Polyketide_cyc2	Polyketide cyclase/dehydrase and lipid transport (PF10604)
GRMZM2G062031	Microtub_bd	Microtubule binding (PF16796)
GRMZM2G128315	LRRNT_2	Leucine rich repeat N-terminal domain (PF08263)
GRMZM2G005984	PSII_Pbs27	Photosystem II Pbs27 (PF13326)
AC191071.3_FG001	FMO-like	Flavin-binding monooxygenase-like (PF00743)
GRMZM2G028643	Malectin_like, LRR_1	Malectin-like domain (PF00560); Leucine rich repeat (PF00560)
GRMZM2G178880	Glyco_trans_2_3	Glycosyl transferase family group 2 (PF13632)
GRMZM2G330907	LRRNT_2, LRR_8, LRR_1, LRR_8, Pkinase	Leucine rich repeat N-terminal domain (PF08263); Leucine rich repeat (PF13855); Leucine rich repeat (PF00560); Leucine rich repeat (PF13855); Protein kinase domain (PF00069)

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
