# Peer review of "Identification and Validation of Candidate Genes Conferring Resistance to Downy Mildew in Maize (Zea mays L.)"

_genes, 2020, doi:10.3390/genes11020191_

Round 1

Reviewer 1 Report

In the manuscript “Identification and validation of candidate genes conferring resistance to downy mildew in maize (Zea mays L.)” the authors were able to identify downy mildew resistant QTLs on 4 out of 10 chromosomes using different strains of downy mildew. Further, candidate genes in the flanked region were identified and validated using qRT-PCR. This study is unique of its type and takes QTL research a step further to look for candidate genes conferring resistance to downy mildew. The reviewer appreciates the efforts that have been put to explore all the candidate genes in the mapped regions including in other species. However, as the authors mentioned, these QTLs were identified previously along with many of the flanking markers, which is the downside of this study. Even though validating candidate genes is an important part of the research, the reviewer thinks authors need to address the following issue before accepting to publish in the journal of Genes. This paper can only be accepted if the following concerns are well. addressed.

Major comments:

In the study, six weeks old resistant genotypes showed consistent phenotypes, however, there is a wide range of variation in gene expression among them. Please explain the possible reasons/s behind it. Only 1/3rd of the SSR markers are polymorphic, which seems less. Please explain the source of SSR markers and the possible reason for reduced transferability. The authors mentioned, there were 3 strains of DM isolate for phenotyping. However, there is a single phenotypic distribution in the mapping population. Is there any strains specific response in the parents and progenies? If so, please explain. Why the phenotype is skewed right? Table 3, please write which allele is contributing to downy mildew resistant rather than resistant genotype. There are set of genes (~17) that are not expressed in either of the phenotypes, why? The authors mentioned the genetic map constructed here is in agreement with most of the published genetic maps using SSR and SNP markers. How do we know that? Is there a possibility to develop a synteny map with the reference to check if there is any recombination? Since the percentage phenotyping response is very low, do you think this study is useful for marker-assisted selection? What other approaches should breeder adopt to gain better resistance, please explain.

Grammar and Punctuation corrections:

Check fonts and spacing throughout the document

Table1: remove information in parenthesis, as it looks crowded

There are many tables in the text, which is overwhelming. Please check, if some of them could be moved to supplementary.

Reviewer 2 Report

The manuscript deals with the identification of QTLs associated with resistance to some downy mildew agents in maize. The manuscript in general is well written and reports very interesting information, since tha authors decided to investigate resistance to multiple causal agents in maize. However, there are some issues that should be fixed .

General comments

The manuscript should be thoroughly revised for what concerns the plant pathology terminology. In the Introduction section, but also in the following sections, confusion is made between the disease (downy mildew) and the pathogens causing it (Perosclerospora and Sclerophthora species). Please note that the plant is not resistant to the disease but to the pathogen(s) causing it. The authors name the causal agents of the disease as “DM strains (Java, sorghum and crazy top)”: since DM is caused by different Perosclerospora and Sclerophthora species, the authors should name the pathogens they’re investigating with their correct species name (i.e. Perosclerospora sorghi, Perosclerospora maydis and Sclerophthora macrospora) in the text. Please, revise the manuscript taking into account this information.

In the Materials and Methods section important information is lacking: at paragraph 2.4 is reported the use of SSR and RFLP markers but their sequences are reported. Please remember that what you write in a manuscript should be reproducible by other authors, therefore all the information must be supplied.

For SSR evaluation, it is reported an agarose gel identification: this means that small size differences could not be evaluated. This should be appropriately discussed.

It is not clear which phenotype data you used for QTL identification (all the data of 2017 season or only the last season ones): can you specify it?

In the results section, lines 189-191 it is written “In this study, the DM-resistant genotypes (CML228, Ki3, and Ki11) and DM-susceptible genotypes (B73 and CML270) studied in April and September 2015 in Phnom Penh, Cambodia are phenotypically similar to DM (Table 1).”: this makes no sense. How can a plant genotype be similar to a disease (DM)? Please rephrase.

In the discussion section, it is not clear the meaning of “Although a QTL analysis is generally necessary for normally distributed phenotype data, it does allow some traits and progenies to be utilized” at lines 326-327.

At lines 345-346 it is written “The QTL regions found in the present study were similar to those in previous studies”: it would be interesting to briefly discuss these similarities, pointing out which DM agents they refere to.

In the discussion section, the literature is quite old. Some fresh studies could be added (e.g. for WRKY transcription factors in grapevine-Plasmopara viticola pathosystem, please see https://www.nature.com/articles/s41598-018-30413-w).

Lines 434-437 “The downregulation of  some of the DM-resistance genes occurs in DM-infected plants. Moreover, it was shown that 30  genes are partially upregulated in DM-infected plants. From these results, we predict that the  phenotype of DM-resistant plants will show resistance and moderate resistance to DM.”: could you please clarify better this concept? It is not clear the conclusion and how you draw it.

Specific comments

Line 38, please change “infection by downy mildew” with “infection by downy mildew agents”.

Line 41, please change “DM is an obligate pathogen” with “DM is caused by obligate pathogens”.

Table 2 legend, pleae explain what “mean” refers to (downy mildew incidence?).

Line 388, please change “Verticillium dahlia” with “Verticillium dahliae” (remember that scientific names must be written in italics).

Line 391, please change “A. thaliana” with “Arabidopsis thaliana” (in italics) and add the name of the discoverer (also for other organisms in the text).

Round 2

Reviewer 2 Report

The authors improved the manuscript according to the referees suggestions and I think that the manuscript is now suitable for publication.